# Serological Response to BNT162b2 and ChAdOx1 nCoV-19 Vaccines in Patients with Inflammatory Bowel Disease on Biologic Therapies

**DOI:** 10.3390/vaccines9121471

**Published:** 2021-12-13

**Authors:** Mohammad Shehab, Fatema Alrashed, Ahmad Alfadhli, Khazna Alotaibi, Abdullah Alsahli, Hussain Mohammad, Preethi Cherian, Irina Al-Khairi, Thangavel Alphonse Thanaraj, Arshad Channanath, Hamad Ali, Mohamed Abu-Farha, Jehad Abubaker, Fahd Al-Mulla

**Affiliations:** 1Division of Gastroenterology, Department of Internal Medicine, Mubarak Alkabeer University Hospital, Kuwait University, Aljabreyah 47060, Kuwait; ahmadalfadhli@moh.gov.kw (A.A.); Khaznanto@agu.edu.bh (K.A.); abd0180037@ju.edu.jo (A.A.); Hussainmohammad@rcsi.com (H.M.); 2Department of Pharmacy Practice, Faculty of Pharmacy, Health Sciences Center (HSC), Kuwait University, Jabriya 13110, Kuwait; falra1@stu.mcphs.edu; 3Department of Biochemistry and Molecular Biology, Dasman Diabetes Institute (DDI), Dasman 15462, Kuwait; preethi.cherian@dasmaninstitute.org (P.C.); irina.alkhairi@dasmaninstitute.org (I.A.-K.); mohamed.abufarha@dasmaninstitute.org (M.A.-F.); jehad.abubakr@dasmaninstitute.org (J.A.); 4Department of Genetics and Bioinformatics, Dasman Diabetes Institute (DDI), Dasman 15462, Kuwait; alphonse.thangavel@dasmaninstitute.org (T.A.T.); arshad.channanath@dasmaninstitute.org (A.C.); hamad.ali@dasmaninstitute.org (H.A.); 5Department of Medical Laboratory Sciences, Faculty of Allied Health Sciences, Health Sciences Center (HSC), Kuwait University, Jabriya 13110, Kuwait

**Keywords:** IBD, COVID-19, vaccine, immunogenicity, infliximab, adalimumab, vedolizumbab, ustekinumab, biologic therapies

## Abstract

**Introduction:** The immunogenicity of SARS-CoV-2 vaccines in patients with inflammatory bowel disease (IBD) on biologic therapies is not well studied. The goal of this study was to measure the serological response to BNT162b2 and ChAdOx1 nCoV-19 vaccines in patients with IBD receiving different biologic therapies. **Methods:** We performed a multi-center prospective study between 1 August 2021 and 15 September 2021. We measured the seropositivity of SARS-CoV-2 antibodies (SARS-CoV-2 IgG) and neutralizing antibody concentrations in patients with IBD receiving biologic therapies 4–10 weeks after their second dose or 3–6 weeks after their first dose of BNT162b2 or ChAdOx1 nCoV-19 vaccines. **Results:** A total of 126 patients were enrolled (mean age, 31 years; 60% male; 71% Crohn’s disease, 29% ulcerative colitis). Of these, 92 patients were vaccinated with the BNT162b2 vaccine (73%) and 34 patients with the ChAdOx1 nCoV-19 vaccine (27%). In patients being treated with infliximab and adalimumab, the proportion of patients who achieved positive anti-SARS-CoV-2 IgG antibody levels after receiving two doses of the vaccine were 44 out of 59 patients (74.5%) and 13 out of 16 patients (81.2%), respectively. In contrast, of those receiving ustekinumab and vedolizumab, the proportion of patients who achieved positive anti-SARS-CoV-2 IgG antibody levels after receiving two doses of the vaccine were 100% and 92.8%, respectively. In patients receiving infliximab and adalimumab, the proportion of patients who had positive anti-SARS-CoV-2 neutralizing antibody levels after two-dose vaccination was 40 out of 59 patients (67.7%) and 14 out 16 patients (87.5%), respectively. On the other hand, the proportion of patients who had positive anti-SARS-CoV-2 neutralizing antibody levels were 12 out of 13 patients (92.3%) and 13 out of 14 patients (92.8%) in patients receiving ustekinumab and vedolizumab, respectively. **Conclusions:** The majority of patients with IBD who were on infliximab, adalimumab, and vedolizumab seroconverted after two doses of SARS-CoV-2 vaccination. All patients on ustekinumab seroconverted after two doses of SARS-CoV-2 vaccine. The BNT162b2 and ChAdOx1 nCoV-19 SARS-CoV-2 vaccines are both likely to be effective after two doses in patients with IBD on biologics. Larger follow-up studies are needed to evaluate if decay of antibodies occurs over time.

## 1. Introduction

COVID-19 has rapidly become a major health concern worldwide. The disease is caused by a new virus called severe acute respiratory syndrome coronavirus 2 (SARS-CoV-2) [1]. To curb the ongoing pandemic caused by SARS-CoV-2 infection, development of vaccines has been advancing at an unprecedented pace, and many countries have authorized different vaccines. BNT162b2 (Pfizer/BioNTech) and the adenovirus vector vaccine ChAdOx1 nCoV-19 (Oxford/AstraZeneca) are commonly used worldwide to face further surges of SARS-CoV-2 infection. Both vaccines have demonstrated effectiveness in preventing severe COVID-19 in a phase III placebo-controlled randomized clinical trial and in real-world data [2,3,4].

The most significant development in the treatment of inflammatory bowel disease (IBD) has been the introduction of anti-TNF-alpha monoclonal antibodies, such as infliximab and adalimumab, and subsequent other new biologic therapies including ustekinumab, an interleukin (IL)-12 and IL-23 inhibitor, and vedolizumab, an integrin antagonist. These medications are approved for moderate to severe ulcerative colitis and Crohn’s disease [5,6]. Patients with IBD treated with biologic therapies are considered to be immunocompromised and may be at increased risk of infection [7,8]. However, in patients with IBD who are receiving biologic therapies, no association has been found with severe COVID-19 outcomes, such as hospitalization and death [9]. In addition, the International Organization for the Study of Inflammatory Bowel Disease (IOIBD) recommends that patients with IBD should be vaccinated against COVID-19 and that vaccination should not be deferred in patients receiving biologic therapy [10]. However, clinical trials excluded patients taking immunosuppressive medications; therefore, it is imperative to evaluate the effectiveness of coronavirus disease 2019 (COVID-19) vaccination in patients with IBD with diverse exposure to biologic therapies. Thus, the goal of this study is to determine the immunogenicity of BNT162b2 and ChAdOx1 nCoV-19 vaccines in patients with IBD receiving different biologic therapies.

## 2. Materials and Methods

We performed a prospective multi-center cohort study at two tertiary care centers (Muabark Alkabeer Hospital in Jabriya and Dasman Center in Kuwait City). Patients were recruited at their time of attendance to the gastroenterology infusion rooms and clinics between 1 August 2021 and 15 September 2021. Our patient inclusion criteria were the following: (1) patients with a confirmed diagnosis of inflammatory bowel disease before the start of the study; (2) patients receiving the tumor necrosis factor antagonists (anti-TNFs) vedolizumab or ustekinumab for at least 6 weeks for induction of remission, or before the next scheduled dose for patients on maintenance therapy (within dose intervals); (3) patients who had received one or two doses of COVID-19 vaccination with the BNT162b2 or ChAdOx1 nCoV-19 vaccine; (4) patients who had received two doses of vaccine within 4–10 weeks before recruitment or received one dose of vaccine within 3–6 weeks before recruitment; and (5) patients who were at least 18 years of age or older. Standard induction and maintenance doses were included (weight-based doses for infliximab and fixed doses for vedolizumab, adalimumab, and ustekinumab). Examples of common standard doses are as follows: infliximab 5 mg/kg every 8 weeks, vedolizumab 300 mg every 8 weeks, adalimumab 40 mg every 2 weeks, and ustekinumab 90 mg every 8 weeks.

Patients were excluded if they had tested positive for SARS-CoV-2 previously or had symptoms of COVID-19 since the start of the pandemic. In addition, patients who received corticosteroids two weeks before the first dose of the vaccine up to the time of recruitment were excluded. Additionally, patients who missed their scheduled biologic dose were excluded. Finally, patients taking immunomodulators such as azathioprine, methotrexate, or other immunosuppressive therapies were also excluded. This study was performed and reported in accordance with the Strengthening the Reporting of Observational Studies in Epidemiology (STROBE) statement (Appendix A) [11].

The diagnosis of IBD was made according to the international classification of diseases (ICD-10 version: 2016). Patients were considered to have IBD when they had ICD-10 K50, K50.1, K50.8, or K50.9, corresponding to Crohn’s disease (CD), and ICD-10 K51, k51.0, k51.2, k51.3, k51.5, k51.8, or k51.9 corresponding to ulcerative colitis (UC) [12].

To evaluate the serological response to BNT162b2 or ChAdOx1 nCoV-19 vaccines in patients with IBD on biologic therapies, we measured the levels of SARS-CoV-2 immunoglobulin G (IgG) and neutralizing antibodies post vaccination. In addition, we calculated the percentage of patients who achieved positive SARS-CoV-2 antibodies levels.

We performed descriptive statistics to present the demographic characteristics of patients included in this study and their measured antibody levels. Analysis was conducted using R (R Core Team, 2017). SARS-CoV-2 antibody levels are expressed as medians with interquartile range (IQR) unless otherwise indicated. In addition, percentages of positive IgG and neutralizing antibody levels post-vaccination were calculated for each biologic therapy (infliximab, adalimumab, vedolizumab, and ustekinumab).

### 2.1. Laboratory Methods

In this study, to assess the magnitude of systemic virus-specific antibodies, anti-SARS-CoV-2 antibodies were quantified in plasma using an enzyme-linked immunosorbent assay (ELISA) kit (SERION ELISA agile SARS-CoV-2 IgG SERION Diagnostics, Würzburg, Germany) based on the manufacturer’s protocol. Units of IgG levels were reported as binding antibody units (BAU)/mL. Values below 31.5 BAU/mL were considered negative or non-protective. Neutralizing antibody levels below 20% were considered negative or non-protective. The positive and negative thresholds were determined as per the manufacturer’s instructions. The results were construed by calculating the inhibition rates for samples as per the following equation: Inhibition = (1 − O.D. value of sample/O.D. value of negative control) × 100%.

### 2.2. Ethical Consideration and Role of Funders

This study was reviewed and approved by the Ethical Review Board of Mubarak Alkabeer Hospital and Dasman Center “Protocol # RA HM-2021-008” as per the updated guidelines of the Declaration of Helsinki (64th WMA General Assembly, Fortaleza, Brazil, October 2013) and of the US Federal Policy for the Protection of Human Subjects. The study was also approved by the reginal health authority (reference: 3799, protocol number 1729/2021). Subsequently, the patients’ informed written consent was obtained before inclusion in the study.

## 3. Results

### 3.1. Patient Characteristics

This study included 126 participants, the mean age was 31 years, and 60% were males. Mean BMI was 25 kg/m^2^. The proportion of patients with Crohn’s disease and ulcerative colitis were 71% and 29%, respectively (see Table 1).

In terms of biologic therapies, 78 (62%) patients were on infliximab and 18 (14%) patients were on adalimumab, whereas the number of patients receiving vedolizumab or ustekinumab was similar at 15 (12%).

A total of 92 patients were vaccinated with BNT162b2 (73%) and 34 patients were vaccinated with ChAdOx1 nCoV-19 (27%). A total of 102 patients received two doses of SARS-CoV-2 vaccine, BNT162b2 or ChAdOx1 nCoV-19, whereas 24 patients received only one dose.

### 3.2. Serological Response to Both BNT162b2 and ChAdOx1 nCoV-19 Vaccines

The proportion of patients who achieved positive anti-SARS-CoV-2 IgG antibody levels after receiving two doses of either BNT162b2 or ChAdOx1 nCoV-19 vaccines in patients treated with infliximab and adalimumab were 44 out of 59 patients (74.5%) and 13 out of 16 patients (81.2%), respectively. In contrast, the proportion of patients who achieved positive anti-SARS-CoV-2 IgG antibody levels after receiving two doses of the vaccine in patients treated with ustekinumab and vedolizumab were 100% and 92.8%, respectively. In patients receiving infliximab and adalimumab, the proportion of patients who had positive anti-SARS-CoV-2 neutralizing antibody levels after two-dose vaccination was 41 out of 59 patients (69.4%) and 14 out 16 patients (87.5%), respectively. On the other hand, the proportion of patients who had positive anti-SARS-CoV-2 neutralizing antibody levels were 12 out of 13 patients (92.3%) and 13 out of 14 patients (92.8%) in patients receiving ustekinumab and vedolizumab, respectively.

### 3.3. Serological Response to the BNT162b2 Vaccine

The proportion of participants with positive anti-SARS-CoV-2 IgG levels was 34 out of 43 patients (79%) in patients vaccinated with BNT162b2 and receiving infliximab, whereas the proportion of patients treated with adalimumab who had positive SARS-CoV-2 IgG levels was 12 out of 15 patients (80%). The proportion of patients treated with infliximab and adalimumab who had positive anti-SARS-CoV-2 neutralizing antibody levels was 30 out of 43 (69.7%) and 13 out of 15 patients (86.6%), respectively. Finally, the proportion of patients treated with ustekinumab and vedolizumab who had positive anti-SARS-CoV-2 IgG levels was 100% and 90.9%, respectively.

### 3.4. Serological Response to the ChAdOx1 nCoV-19 Vaccine

The proportion of participants with positive anti-SARS-CoV-2 IgG and neutralizing antibody levels was 10 out of 16 (62.5%) in patients vaccinated with the ChAdOx1 nCoV-19 vaccine receiving infliximab. The proportion of patients treated with ustekinumab and vedolizumab who had positive anti-SARS-CoV-2 IgG and neutralizing antibody levels was 100%. Only one patient on adalimumab received the ChAdOx1 nCoV-19 vaccine. The levels of anti-SARS-CoV-2 IgG and neutralizing antibodies were above the positive threshold for that patient.

### 3.5. Serological Response to One Dose of Either the BNT162b2 or ChAdOx1 nCoV-19 Vaccine

In total, 24 patients of our cohort received a single dose of BNT162b2 or ChAdOx1 nCoV-19 vaccines. Of those, 18 (75%) patients had positive anti-SARS-CoV-2 IgG and neutralizing antibody levels 3–6 weeks after first dose. Out of the total patients who received a single dose of the vaccine, 19 were on infliximab, two patients were on adalimumab, two were on ustekinumab, and one was on vedolizumab. In patients receiving infliximab, 14 out of 19 (73.6%) had positive anti-SARS-CoV-2 IgG and neutralizing antibody levels.

### 3.6. Median SARS-CoV-2 IgG and Neutralizing Antibody Levels

The median (interquartile range (IQR)) anti-SARS-CoV-2 IgG level was 103 BAU/mL (38, 181) in patients who received two doses of the BNT162b2 vaccine and were treated with infliximab, whereas the median was 90 BAU/mL (48, 134) in patients receiving adalimumab. Respective anti-SARS-CoV-2 IgG levels were 130 BAU/mL (104, 159) and 139 BAU/mL (117, 180) in patients treated with ustekinumab and vedolizumab (Figure 1). Neutralizing antibody levels were 76 (17, 95) and 59 (32, 71) in patients treated with infliximab and adalimumab, respectively. Neutralizing antibody levels in ustekinumab and vedolizumab users were 84 (69, 88) and 88 (79, 95), respectively. (See Table 2)

The median (IQR) anti-SARS-CoV-2 IgG level was 65 BAU/mL (14, 131) in patients who received two doses of the ChAdOx1 nCoV-19 vaccine and were treated with infliximab. The respective anti-SARS-CoV-2 IgG levels were 114 BAU/mL (69, 120) and 120 BAU/mL (88, 131) in patients treated with ustekinumab and vedolizumab. Median neutralizing antibody levels were 49 (10, 84) in patients treated with infliximab (Figure 2). Median neutralizing antibody levels in ustekinumab and vedolizumab users were 82 (37, 86) and 61 (42, 75), respectively (see Table 3). Finally, in patients who received one dose, the median (IQR) anti-SARS-CoV-2 IgG and neutralizing antibody levels were 60 (33, 151) and 34 (20, 93), respectively (See Table 4).

## 4. Discussion

The serological response to one or two doses of BNT162b2 and ChAdOx1 nCoV-19 vaccines was evaluated in a cohort of 126 patients with IBD receiving different biologic therapies. Most patients seroconverted and had positive antibody responses 4–10 weeks after the second vaccine dose. Interestingly, while the majority of patients (70–90%) on anti-TNFs or vedolizumab achieved positive anti-SARS-CoV-2 IgG and neutralizing antibody levels, all patients receiving ustekinumab achieved positive anti-SARS-CoV-2 antibody levels regardless of the vaccine used.

Our findings reinforce the results of previous studies indicating positive humoral immune response with complete vaccination in patients with IBD being treated with biologic therapies. Deepak et al. found that in a cohort of 133 participants with different chronic inflammatory conditions and receiving immunosuppressive therapy, 88.7% had detectable anti-SARS-CoV-2 antibody levels after two doses of the BNT162b2 or mRNA-1273 vaccine. However, the mean concentration of anti-SARS-CoV-2 antibodies in patients with inflammatory disease was lower compared to immunocompetent participants [13].

Our results showing seroconversion across medication groups are consistent with other IBD studies. One study [14] assessed humoral responses after mRNA vaccination in adults with IBD at different time points. The study found that, regardless of medication regimen, 99% of participants had detectable antibodies after 2 weeks. In addition, anti-SARS-CoV-2 IgG levels were lowest in patients receiving anti–tumor necrosis factor combination therapy or corticosteroids. However, it is important to mention that the study was not powered to assess differences between medication subgroups.

Khan et al. explored the effectiveness of mRNA vaccination in veteran patients with IBD receiving diverse immunosuppressive medications [15]. The authors observed that two doses of an mRNA vaccine, but not a single dose, was associated with 69% reduced hazard of infection relative to an unvaccinated status. They also suggested that SARS-CoV-2 vaccine effectiveness is similar in patients with IBD regardless of the immunosuppressive agents being used. Wong et al. reported serological responses to mRNA vaccination in patients with IBD on biologic therapies [16]. They found that despite achieving antibody levels consistent with presumed protection, there was an association of lower antibody levels in patients with vedolizumab for all antibodies tested and with anti-TNFs for the receptor binding domain (RBD) of the SARS-CoV-2 S protein total immunoglobulin.

One study [17] reported the effectiveness of the BNT162b2 mRNA COVID-19 vaccine in patients with IBD receiving different therapies, including biologic. The authors found that in patients with IBD, the vaccine was highly effective and comparable to the general population with an absolute breakthrough infection rate of 0.1% for fully vaccinated IBD patients. However, the study suggested that patients with Crohn’s disease (CD) may have an increased risk for breakthrough infections, but larger studies are needed to validate this finding.

Anti-SARS-CoV-2 circulating antibodies play a major role in risk reduction of severe COVID-19. The IgG class particularly targets the spike protein of SARS-CoV-2, via its S1 subunit or its receptor binding domain (RBD) [18]. Thus, the binding of the virus to host receptors is weakened or terminated completely. Neutralizing antibodies are major contributors to immunity [19,20]. Vaccine studies have shown that high levels of IgG and neutralizing antibodies against the SARS-CoV-2 spike protein correlate with protection. However, it is not currently known what the threshold titers should be to achieve protection.

The CLARITY IBD study [21] showed that the lowest rates of seroconversion were observed in participants treated with infliximab in combination with an immunomodulator with both the BNT162b2 or ChAdOx1 nCoV-19 vaccines. Interestingly, the highest rates of seroconversion were seen in patients treated with vedolizumab monotherapy who received either one of the previously mentioned vaccines.

Anti-TNF agents inactivate the proinflammatory cytokine TNF by direct neutralization, thus resulting in suppression of inflammation. The cytokine TNF is involved in multiple aspects of host immune responses, including T-cell dependent antibody production. This mechanism of action may in part explain why TNF blockade is clinically beneficial, but also explain the increased risk of serious and opportunistic infections and impaired response to some vaccines [22]. The mechanism of action of vedolizumab, an anti-integrin, is hypothesized to be restricted to the gastrointestinal tract, not affecting immune responses outside of the gastrointestinal tract [23]. Finally, ustekinumab inhibits the p40 subunit of interleukin (IL)-12/23, which are major drivers of the adaptive immune response [24].

Taken together, these emerging data provide reassurance that biologic therapies do not markedly reduce the response to COVID-19 immunization and support recent consensus recommendations to vaccinate all patients with IBD regardless of immune-modifying therapies [25]. However, patients should be advised that vaccine efficacy may be decreased when receiving systemic corticosteroids. In addition, anti-SARS-CoV-2 antibody decay remains a legitimate concern in patients with IBD on biologic therapy [26]; therefore, booster doses are being recommended by official organizations [27,28].

Our study has several strengths. It is a multi-center prospective study. It examined the effect of two important SARS-CoV-2 vaccines in a vulnerable patient population with IBD on common biologic agents, who might be reluctant to vaccinate. It is well designed with a low risk of bias, given its inclusion and exclusion criteria. It addresses a very clinically relevant question that physicians and patients are eager to answer.

However, our study has some limitations. It includes a small sample size of patients; thus, it was not powered to assess differences across medication subgroups. Furthermore, only humoral serological responses were assessed, and it is likely that T cell-mediated immunity also plays a protective role. Finally, given the observational nature of the study, possible confounders may exist, such as difference in biologic dosing and serum drug concentrations that were not accounted for. Larger follow-up studies are needed to evaluate if decay of antibodies occurs over time and if a possible third booster vaccine dose may be needed.

## 5. Conclusions

Our study showed that most patients with IBD on infliximab, adalimumab, and vedolizumab biologic therapies seroconverted after two doses of SARS-CoV-2 vaccination. All patients on ustekinumab seroconverted after two doses of SARS-CoV-2 vaccine. The BNT162b2 and ChAdOx1 nCoV-19 SARS-CoV-2 vaccines are both likely to be effective after two doses in patients with IBD on biologics. Larger follow-up studies are needed to evaluate if decay of antibodies occurs over time and if a third booster vaccine dose may be needed.

## Figures and Tables

**Figure 1 vaccines-09-01471-f001:**
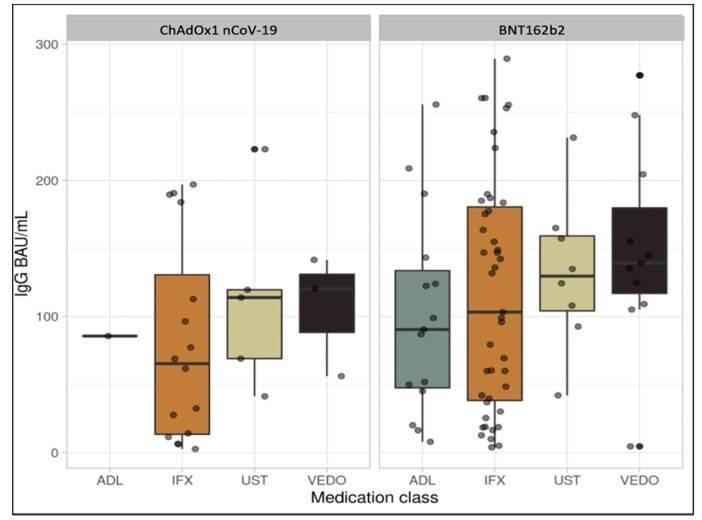
Box plot illustrating IgG antibody concentrations among patients on different biologic therapies.

**Figure 2 vaccines-09-01471-f002:**
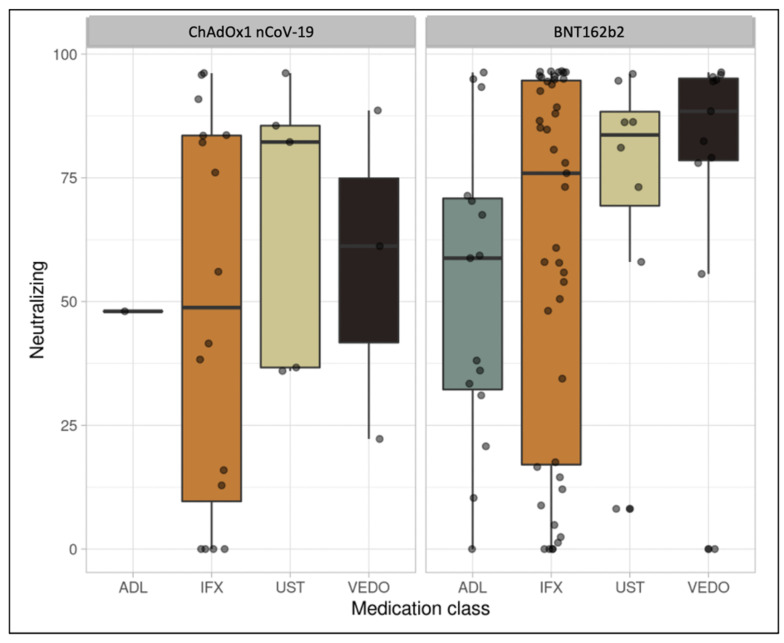
Box plot illustrating neutralizing antibody concentrations among patients on different biologic therapies.

**Table 1 vaccines-09-01471-t001:** Demographic characteristics of all included patients.

Characteristic	Overall, N = 126	Doses	
		One, N = 24	Two, N = 102
Mean age (years)	31 (24, 39)	27 (21, 37)	32 (25, 40)
Gender			
Female	51 (40%)	8 (33%)	43 (42%)
Male	75 (60%)	16 (67%)	59 (58%)
BMI	25 (21, 29)	24 (21, 28)	25 (21, 29)
Disease extent, n (%)			
Ulcerative colitis (UC)	36 (29.0%)	7 (29.1%)	31 (30.3%)
E1: ulcerative proctitis	7 (19.4%)	1 (14.2%)	7 (22.5%)
E2: left sided colitis	11 (30.5%)	3 (42.8%)	10 (32.2%)
E3: extensive colitis	18 (50.0%)	4 (57.1%)	14 (45.1%)
Crohn’s disease (CD)	90 (71.0%)	17 (70.9%)	71 (69.7%)
L1: ileal	40 (44.4%)	9 (52.9%)	38 (53.5%)
L2: colonic	15 (16.6%)	2 (11.7%)	12 (16.9%)
L3: ileocolonic	31 (34.4%)	5 (29.4%)	19 (26.7%)
L4: upper gastrointestinal	4 (4.4%)	1 (5.8%)	2 (2.8%)
B1: inflammatory	44 (48.8%)	10 (58.8%)	37 (52.1%)
B2: stricturing	21 (23.3%)	2 (11.7%)	14 (19.7%)
B3: penetrating	25 (27.7%)	5 (29.4%)	20 (28.1%)
Lab Parameters			
CRP, mg/L (median)	6.1	6	6.5
Albumin, g/L (median)	40	41	40
Stool fecal calprotectin, ug/g	112	112	110
(median)			
ESR, mm/hr	9	9	8
Co-morbidities			
Diabetes	5 (4.0%)	1 (4.2%)	4 (3.9%)
OSA	1 (0.8%)	0 (0%)	1 (1.0%)
Hypertension	2 (1.6%)	0 (0%)	2 (2.0%)
Heart disease	2 (1.6%)	0 (0%)	2 (2.0%)
Arthritis	5 (4.0%)	0 (0%)	5 (4.9%)
COPD	1 (0.8%)	0 (0%)	1 (1.0%)
Kidney	2 (1.6%)	0 (0%)	2 (2.0%)
Asthma	8 (6.3%)	0 (0%)	8 (7.8%)
Hyperlipidemia	4 (3.2%)	1 (4.2%)	3 (2.9%)
Vaccine type			
ChAdOx1 nCoV-19	34 (27%)	9 (38%)	25 (25%)
BNT162b2	92 (73%)	15 (62%)	77 (75%)
Median anti-SARS-CoV-2 antibody levels (IQR)			
IgG BAU/mL	101 (40, 165)	60 (33, 151)	109 (43, 165)
Neutralizing	64 (25, 93)	34 (20, 93)	73 (34, 93)
Number of patients on biologic therapy n(%)			
Adalimumab	18 (14%)	2 (8.3%)	16 (16%)
Infliximab	78 (62%)	19 (79%)	59 (58%)
Ustekinumab	15 (12%)	2 (8.3%)	13 (13%)
Vedolizumab	15 (12%)	1 (4.2%)	14 (14%)
Median duration of biologic therapy use (months)			
Adalimumab	12	12	12
Infliximab	13	12	13
Ustekinumab	12	11	12
Vedolizumab	10	10	10

**Table 2 vaccines-09-01471-t002:** Serological response in participants vaccinated with the BNT162b2 vaccine and receiving biologic therapies.

Characteristic	Overall N = 77 ^1^	ADL, N = 15 ^1^	IFX, N = 43 ^1^	UST, N = 8 ^1^	VEDO N = 11 ^1^
IgG BAU/mL	124 (49, 175)	90 (48, 134)	103 (38, 181)	130 (104, 159)	139 (117, 180)
Neutralizing	76 (34, 94)	59 (32, 71)	76 (17, 95)	84 (69, 88)	88 (79, 95)

^1^ Median (IQR) or frequency (%).

**Table 3 vaccines-09-01471-t003:** Serological response in participants vaccinated with ChAdOx1 nCoV-19 and receiving biologic therapies.

Characteristic	Overall N = 25 ^1^	ADL, N = 1 ^1^	IFX, N = 16 ^1^	UST, N = 5 ^1^	VEDO, N = 3 ^1^
IgG BAU/mL	77 (32, 120)	86 (86, 86)	65 (14, 131)	114 (69, 120)	120 (88, 131)
Neutralizing	56 (22, 84)	48 (48, 48)	49 (10, 84)	82 (37, 86)	61 (42, 75)

^1^ Median (IQR) or frequency (%).

**Table 4 vaccines-09-01471-t004:** Serological response among patients who had received only one dose of either the BNT162b2 or ChAdOx1 nCoV-19 vaccine.

Characteristic	Overall N = 24 ^1^	ADL, N = 2 ^1^	IFX, N = 19 ^1^	UST, N = 2 ^1^	VEDO, N = 1 ^1^
IgG BAU/mL	60 (33, 151)	93 (67, 118)	68 (32, 153)	27 (23, 31)	197 (197, 197)
Neutralizing	34 (20, 93)	27 (22, 32)	44 (20, 94)	29 (29, 30)	94 (94, 94)

^1^ Median (IQR) or frequency (%).

## Data Availability

All data are available within the manuscript.

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
