# Peer review of "Serological Response to BNT162b2 and ChAdOx1 nCoV-19 Vaccines in Patients with Inflammatory Bowel Disease on Biologic Therapies"

_vaccines, 2021, doi:10.3390/vaccines9121471_

Round 1

Reviewer 1 Report

Dear all the listed authors,

 Delta variant has been reported first in India, but the type of mutation was reported in variants such as Beta that emerged earlier. On August 14, 2021, WHO announced the Solidarity PLUS trial with three therapies: artesunate, imatinib and infliximab.  Infliximab is recommended for diseases of the immune system such as Crohn’s Disease and rheumatoid arthritis.

After reading through the submission, the manuscript disseminated few knowledge for patients and researchers due to poor experimental design  and small sample size and limited innovation. Please find the detailed comments for your improvement:

  1. No specific doses and prolonged time of medications, infliximab, adalimumab,ustekinumab and vedolizumab.As for two vaccines against COVID-19, BNT162b2 is mRNA vaccine while ChAdOx1n CoV-19 is a vector vaccine, they are unsuitable for the control vaccine in light of different humoral immune response.
  2.  In Table 1, it is very complicated to understand the experimental protocol. Also, authors did not mention how to detect decay of antibody levels post treatment with above anti-TNF antibodies.
  3. The conclusion "All patients on ustekinumab seroconverted after two dose doses of SARS-CoV-2 vaccine"was based on only 2 patients and it  is required to modify the statement both in Abstract and Conclusion sections.
  4. It is not clear immune responses in light of disease extent(UC and CD) and co-morbidities post treatment with above 4 antibodies.

Author Response

Please find attached the reply to your comments. Thank you.

Reviewer 2 Report

This is an important study investigating the effects of IBD treatment on the immunogenicity of COVID vaccination.  Although it is a small pilot study, it is well designed and controlled, and one of the few studies looking at specific immunocompromised groups. The data is presented clearly and the paper is well written.

I just have a few minor comments:

  • it would be helpful to include the city and country in which the trial was undertaken
  • please mention what was used for the neutralization assay - was it live virus or a pseudovirus?  Was a TCID50 or plaque assay used?

Author Response

Please find the reply to your comments. Thank you.

Reviewer 3 Report

Since December 2019, the entire world is still fighting against an infection caused by a Coronavirus, designated as SARS-CoV-2. Actually, the vaccination campaign is the first method to counteract the COVID-19 pandemic; however, sufficient vaccination coverage and vaccine acceptance need to studied. In this context, the paper under review is aimed at measuring serological response to BNT162b2 and ChAdOx1 vaccines in patients with Inflammatory Bowel Disease on Biologic Therapies.

Although, the issue is interesting, at this stage, the enrolled sample is not enough to support the conclusion provided. I suggest to submit the paper to a minor journal as pilot study.

I hope that these comments will not discourage the authors to deep the investigation on this field and on the selected issue.

Author Response

(The authors gave the same response as above.)

Reviewer 4 Report

The authors present a study on the serological response (serum antibody levels) in IBD patients receiving Covid-19 vaccinations following 4 types of biologic therapies. This paper is fairly well written, but has some incorrect structural problems associated with certain components not being reported in the correct locations and in the proper format to achieve an effective presentation.  There are also a number of wording issues and important omissions of information that need to be corrected as follows:

Title

suggested title revision - remove the ending words: ; A Multi-Center Prospective Study which is not necessary and distracts from a simple clear title (without a run-on phrase that does not contribute to the main theme) 

Key words: add biologic therapies

Introduction

Line 58 - revise wording: therapy should be either: 1) a biologic therapy or 2) biologic therapies (plural)

Line 59 - define biologic therapies including the various kinds available with references. Introduce the 4 specific types (ADL, IFX, UST, and VEDO) studied in this paper and what they are most used for (therapeutic applications).

Materials and Methods

Line 71 - move Lines 126-135 (Patients' Characteristics from Results) and insert here at the beginning of the Materials and Methods followed by Table 1 (which are not study results, but pre-study patient history and characteristics).

Line 74 add words: our patient inclusion criteria are the following:

Line 83 - remove space before the word "received"

Line 103 - add the word therapy after biologic

Results

Each of the following 5 paragraphs is begun with a topic sentence consisting of an indirect statement, followed by a comma, e.g. In patients,.....

These topic sentences (beginning new paragraphs) as much stronger, and more effective as direct statements, such as (an example):

The proportion of participants with xx....., vaccinated with xx and receiving xx biologic therapy, was xx out of xx patients (xx%), ...

This need for a direct statement applies for paragraphs beginning with the following sentences: L 150, L 159, L 167, L 174, and L 183.

Lines 173-190 - The text in this section needs to be parsed into individual paragraph(s) that each introduce a single Table or Figure (with some following explanatory statements about key results) reported in each. Each paragraph should be followed immediately with the Table or Figure introduced. As currently structured, all of the Tables and Figures are bunched together in sequence without any explanatory text preceding each data presentation. It is much easier to follow results when parsed with explanatory text before each new data is presented and then provide qualifying statements comparing results based on treatment differences and effects on serum antibodies.

Table 2 and Table 3 - reword titles:

Serological response in participants vaccinated with XX vaccine and receiving biologic treatments. 

Placing Biological Treatments with a line over the 4 types (within Tables 2, 3, and 4) would improve clarity

Discussion

Line 212 (indirect statement in topic sentence) - move the preposition at the beginning of the sentence (followed by a comma) to the end of the sentence to make a direct statement.

Line 226 reword - Our results showing seroconversion across...

Conclusions

Line 292-293 add words biologic therapies after word "vedolizumab"

Line 296 rephrase:

Larger follow up studies are needed to evaluate if decay of antibodies
occurs over time and if a third booster vaccine dose may be needed.

Author Contribution -

use normal font size for author's names

Author Response

(The authors gave the same response as above.)

Round 2

Reviewer 1 Report

Dear all the listed authors,

  The resubmission improved after revision and it will disseminate clinical knowledge to potential researchers and doctors facing inflammatory bowel disease based on limited samples. However, authors have to bear in mind that BNT162b2 and ChAdOx1n CoV-19 are unsuitable for the control group due to different immune responses.Particularly,ChAdOx1n COVID-19 is adenovirus-vector vaccine and IgG responses increase gradually and reduced humoral response will be induced after three times.

 The manuscript is recommended to be research report based on the present preliminary data.

   Thank you for your attention.

All the best wishes,

 Reviewer 2#

Author Response

Thank you for this useful feedback. 

Title changed as adviced. Comment added to the method section.

Reviewer 3 Report

I confirm that the issue is interesting; but, at this stage, the enrolled sample is not enough to support the conclusion provided. I suggest to submit the paper to a minor journal as pilot study.

Author Response

Thank you. Comment addressed in the previous revision.

Reviewer 4 Report

The authors have made very significant improvements in the manuscript through all of their diligent revisions in great detail as suggested.  I appreciate the additional work the authors have put into making the modifications required to cover and correct the limitations of the manuscript.  I still feel that Table 1 should be moved to the Materials and Methods because this patient data technically are not results, but pre-experimental parameters set up for the experimental design. Nevertheless, I respect the authors' choice of keeping this information in the Results section.

Author Response

Thank you so much for your kind words. Your comments helped us improve our manuscript.